# Protocol for a scoping review of the evidence concerning the unique needs and experiences of Orthodox Jewish women and their partners using maternity services

**Michal Rosie Meroz**[1,2], **Christine McCourt**[3], **Carol Rivas**[4*]

1 Homerton Healthcare NHS Foundation trust, London, United Kingdom, 2 National Institute for Health and Care Research (NIHR), London, United Kingdom, 3 City St George's, University of London, London, United Kingdom, 4 Social Research Institute, University College London (UCL), London, United Kingdom

* c.rivas@ucl.ac.uk

## Abstract

### Background

Orthodox Jews follow the Jewish law, Halacha, that determines most daily activities and behaviours. Halacha restrictions and the insular lifestyle of groups within the Orthodox community have led to cultural barriers when interacting with NHS maternity services in the UK.

### Aims and objectives

This protocol describes a scoping review that will aim to: explore the needs of Orthodox Jews when interacting with maternity services and their experiences in the UK; to what extend this topic has been studied; and identify any gaps that need further research.

### Rationale

The literature on this topic is scarce. There is an urgent need to understand the unique needs of these communities in order to make NHS maternity services accessible to all.

### Methods

The scoping review will follow Arksey and O'Malley's framework for scoping reviews. We will utilise a broad search strategy that will include terms such as Orthodox Jewish, Haredi, Halacha, Needs, Experiences and Maternity Services. We will search the grey literature and databases such as OpenGrey, PubMed, Web of Science, CINHAL, SocINDEX and ProQuest. The search will be an iterative process that will be led by the search itself and the Patient and Public Involvement (PPI) work done in parallel.

**Data availability statement:** No datasets were generated or analysed during the current study. All relevant data from this study will be made available upon study completion.

**Funding:** NIHR ICA Pre-doctoral Clinical and Practitioner Academic Fellowship 2024 Reference: NIHR304873.

**Competing interests:** The authors have declared that no competing interests exist.

## Inclusion criteria

Using the terms above, we will include English papers from all OECD countries, applying no restrictions on publication year.

## Expected outcomes

Mapping the literature will allow a better understanding of the needs and experiences of the Orthodox community when interacting with NHS maternity services in the UK and will lead to the next stage of the project that aims to make these services more culturally sensitive.

## Introduction

There are approximately 300,000 Jewish people living in the UK today, and while this accounts for only 0.5% of the total population in the country, it is the fifth largest Jewish community in the world [1]. The literature indicates that some Jewish people experience judgement, stereotyping, inequalities, barriers to health services and lack of understanding [1–4]. A recent NHS Race and Health Observatory report stated that Jewish communities in the UK "*have often been neglected and overlooked in conversations about health inequalities*" [5].

The Jewish community is diverse and heterogenous, and while most Jews share the same customs, festivals, values and attachment to Israel, the most distinct group within Judaism is the Orthodox Jewish group, and the strictly Orthodox (Haredi) amongst them, that represent 19% of the total Jewish population in the UK [1].

The Haredi is one of the fastest growing communities in the world [6] due to a high total fertility rate of 7 children per woman (nearly four times that of the general population in the UK) [7]. Most Haredim in the UK live in enclaves in Salford, Manchester and the London boroughs of Hackney and Haringey [5,8].

Members of the Orthodox community follow the Jewish law (*Halacha*), that interprets the Bible's commandments and incorporates them into most aspects of life [6]. One example is the *"be fruitful and multiply and fill the earth"* commandment (Genesis 1:28) that has led to the high fertility rate reported above [9]. Membership in these communities also means adopting an observant lifestyle and Haredi live in a strictly religious environment, separated from the "external" world (outside the community) [6,8]. One of the main restrictions that Orthodox Jewish people follow is honouring the *Shabbat*/*Shabbos* (the Hebrew name for Saturday), which means avoiding any work during this day, including driving, cooking, using electronic devices and writing. *Halachic* laws also mean that any physical contact between people of the opposite sex, even a handshake or a pat on the shoulder, outside the nuclear family, is strictly prohibited, unless medically required; this also means that some Haredi will avoid eye contact with the opposite sex [9]. Amongst Orthodox Jewish women, modesty is highly significant, and marriageability and family are of high value [9,10].

Due to their unique lifestyle, religious beliefs, and related customs, Orthodox Jews have specific health and social needs. Many Haredi were born outside the UK and do

not speak English as their first language [9]. Evidence from around the globe also suggests that Haredi women may be at higher risk of fully or partially declining breast cancer and pregnancy screening tests, despite higher prevalence of specific genetic disorders, such as Tay-Sachs disease and BRCA, amongst the Jewish population [2,10]; other studies have reported low uptake of national children's vaccination programmes amongst Haredim in London [4]. Some Haredi women may decline a caesarean section as it limits the number of future pregnancies [9]. Data collection on health outcomes for the Orthodox population in the UK is limited, as neither the census nor NHS health records routinely record Jewish identity or Orthodox affiliation [5].

There is evidence to suggest that maternity services are sometimes the only interaction between Haredi women and the "external" world [2]. The above-specified restrictions, in combination with adherance to Jewish holidays restrictions and a diet that is based on Kosher food only, has the potential of leading to cultural barriers when interacting with healthcare professionals, as already discussed in the literature [5,9,10]. An insular lifestyle has also resulted in limited literature on the Orthodox Jews in general, and on Orthodox Jewish women's health in particular.

The proposed scoping review aims to focus on one aspect of healthcare, maternity services. Due to the high fertility rate within the community and the sensitive nature of reproductive health and childbirth, this service may be the "hottest" point of interaction between Orthodox Jewish communities and the NHS. The main author is a senior research midwife at Homerton Healthcare NHS Foundation Trust, that provides health services in Hackney, home to the largest Orthodox Jewish community in the UK. A local report from 2011 estimated 7.4% of Hackney residents and 22% of the children in the borough are Haredi [11]. The proposed review will be undertaken as part of the main author's National Institute for Health and Care Research (NIHR) PCAF (Pre-doctoral Clinical and Practitioner Academic Fellowship). The co-authors are academic supervisors who will act as advisors and co-reviewers, adding their depth and breadth of knowledge in the field of social and health sciences. It is anticipated that the results of this scoping review, and the patient and public involvement (PPI) work done in parallel, will inform the development of a Participatory Action Research (PAR) protocol, as part of a doctoral project of the main author. Mapping the existing knowledge, and identifying gaps, will enable a better understanding of what the next step should be to make NHS maternity services more culturally sensitive to the Orthodox Jewish community, or more "*Kosher*".

## Methodology

### Scoping review

According to Grimshaw [12, p.34], scoping reviews are "*exploratory projects that systematically map the literature available on a topic, identifying key concepts, theories, sources of evidence and gaps in the research*". The aim of the proposed review is to identify, explore and map the literature about Haredi women in the UK and their interaction with maternity services using the breadth of Grimshaw's approach [12] and to identify gaps in the literature, therefore a scoping review was identified as the best methodology.

The proposed scoping review will be conducted in line with the methodological framework of Arksey and O'Malley [13] and will follow their proposed stages: identifying the research question, identifying relevant studies, selecting studies, charting the data, and collating, summarizing and reporting the results. We will also follow the updated guidance for conducting scoping reviews by Peters et al. [14]. We will adhere to PRISMA-ScR (PRISMA Extension for Scoping Reviews) and will include the following sections in our review: abstract, introduction, methods, results and discussion. Each of these sections will have sub-sections, as directed by the PRISMA tool [15].

We aim to publish the findings of the review, as Arksey and O'Malley [13] suggested that scoping reviews can be published as a method in their own right.

### Identifying the research question

The research question for the proposed scoping review is: **what is known about the needs of Orthodox women and their partners that are using maternity services?**

The sub-questions:

1. How are these needs being met or not in UK maternity services?

2. What is known about the experiences of Orthodox women and their partners that are using maternity services in the UK?

## Identifying relevant studies

Following Arksey and O'Malley's [13] recommendation, the proposed scoping review will use a broad search strategy and will therefore cover a range of study designs and qualities. This will include peer reviewed papers as well as "grey" literature, such as local reports and community publications (e.g., rabbinical instructions). We will identify studies via different sources such as electronic databases, reference lists of other relevant studies (including backward and forward citation search), hand-searching of key journals and others (for example, brochures/leaflets that will be collected by community members). If we decide to contact authors of published papers for additional data, this will be stated in the review.

As a first step, the following databases will be searched: OpenGrey, PubMed, Web of Science, Cinahl, SocINDEX, Ethos, and ProQuest. However, the search will be an iterative process, informed by the PPI work that will be conducted in parallel to the literature search.

## Selection of eligible studies

The literature search strategy will be as comprehensive as possible within the time and budget constraints of the PCAF bridge programme (six months in total). We will identify and agree on keywords to be used in our search, these will be presented in a table, as demonstrated in Table 1. This, however, will be an iterative process, and as we immerse ourselves in the evidence, additional keywords may be identified and incorporated into the search strategy. The search strategy will be piloted by the main author, to assess the appropriateness of keywords and databases.

The main author, supported by a qualified librarian, will conduct the initial review of titles and abstracts, using the above databases and the agreed keywords. This will be followed by a screening process, conducted by the main author, to remove any obvious irrelevant papers. To assess for consistency, a second reviewer will independently repeat this process with a sample of papers, and the approach will be adjusted if indicated. The main and one co-author will then review papers' abstracts and full texts, to identify relevant studies, against the inclusion criteria. Any disagreement will be resolved by either consensus or by a third reviewer (the other co-author). We will utilise the PRISMA flow diagram to illustrate the literature search process [16].

**Table 1.  search strategy and keywords.**

| | |
|---|---|
| *Research question 1: what is known about the unique needs of Orthodox Jewish women and their partners that are using maternity services?* <br> *Sub-questions:* <br> *what is known about the extent to which the needs of Orthodox Jewish women and their partners that are using maternity services in the UK being met or not?* <br> *what is known about the experiences of Orthodox Jewish women and their partners that are using maternity services in the UK?* | |
| **Search strategy** | **Key words** |
| **P**opulation: Orthodox Jewish women and partners | Haredi* OR Charedi* OR "Orthodox Jewish" OR "Ultra-Orthodox Jewish" OR Hasidi* OR Chasidi* <br> AND <br> Women OR Female* OR Partner* OR Spouse* |
| **C**oncept: Jewish laws, needs, experiences | Need* OR Halacha OR Halakha OR "Jewish law" OR Experience* OR "Patient experience* OR Halacha OR Halakha OR "Jewish law" |
| **C**ontext: maternity services, maternity care | "Maternity service*" OR "Maternity care" OR "Antenatal care" OR "Postnatal care" OR "Intrapartum care" OR Childbirth |

## Inclusion criteria

The inclusion criteria for the proposed scoping review will be predefined and will follow the Population, Concept, Context (PCC) framework, recommended by the Joanna Briggs Institute guidance for Scoping Reviews [14].

**P-** Ort hodox Jewish women and their partners

**C-** needs and experiences, Jewish laws

**C-** maternity services

We will include original papers discussing the experiences and needs of Orthodox Jewish women and/or their partners when interacting with maternity services from conception to one year postpartum. Papers that conducted interviews at a different time postnatally will be included, as long as they investigated the experiences and/or needs within this timeframe.

An initial scoping search we conducted suggested that the literature on the topic is scarce; this led us to make a few decisions, as described here. Firstly, although the Haredi are the most observant group within Orthodoxy, we decided to include all Orthodox Jewish people. This is also because the terms "Orthodox" and "Haredi" are sometimes used interchangeably.

We have also decided to extend the search beyond the UK, to enable us to identify gaps in UK maternity services that may not exist in other countries. To this end the search will include papers in English from all OECD countries, including Israel, the home to the largest Haredi community in the world. We have also decided to include papers from all publication years; this, however, may be amended, depending on the search results.

## Exclusion criteria

To ensure that data included in the review is valid and full we will exclude papers with no available full text. We will also exclude papers that are not specific to Orthodox Jewish people and/or maternity services. Papers in languages other than English will also be excluded due to lack of resources and the assumption that most relevant studies would be from Israel, North America and the UK and in English.

## Charting the data

As recommended in the scoping review literature, the search process will be detailed in a clear and transparent way to allow for replication [13,14]. This will include the search strategy, key terms used and search dates. We will describe any amendments to the protocol in the main text. As stated by Tricco et al [15] no risk of bias assessment is required for scoping reviews.

## Collating, summarising and reporting the results

All team members will independently read the included papers. Following Thomas and Harden's thematic synthesis process [17], the main author will conduct a line-by-line coding and draft initial themes, which will be shared with all team members via email. We will then hold a series of collaborative discussions via Microsoft Teams to explore the content of the studies and assess the extent to which the emerging themes reflect the data. Following these discussions and consensus-building, the main author will summarise the findings in a shared Excel spreadsheet, which will be accessible to all authors. The number of studies identified, the emerging themes, and the main results of the studies will be reported.

Adapting Arksey and O'Malley's recommendation [13], we will create a table, using Excel software, to present the following information for each study: type of paper (e.g., research/ local report), author/organisation affiliation, year of publication, study populations, study aim/focus, geographic location, study design and methodology, and themes and categories of the main findings. Lastly, we will identify and discuss any gaps in knowledge.

## Discussion

The proposed scoping review aims to identify the most relevant literature and gaps regarding the needs and experiences of Orthodox Jewish women and their partners when interacting with maternity services in the UK. The review will be the first stage in a larger project that aims to investigate this topic from different perspectives (service users and healthcare professionals) and inform relevant changes to current practice, if these are needed. To the best of our knowledge, this is the first research project in the UK of its kind.

## Author contributions

**Conceptualization:** Michal Rosie Meroz.

**Data curation:** Michal Rosie Meroz.

**Formal analysis:** Michal Rosie Meroz.

**Funding acquisition:** Michal Rosie Meroz.

**Investigation:** Michal Rosie Meroz.

**Methodology:** Michal Rosie Meroz.

**Project administration:** Michal Rosie Meroz.

**Supervision:** Carol Rivas, Christine McCourt.

**Writing – original draft:** Michal Rosie Meroz.

**Writing – review & editing:** Carol Rivas, Christine McCourt.

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
