## [Decision Letter · Decision Letter 0]

1 Jul 2025

Dear Dr. Rivas,

Thank you for submitting your manuscript to PLOS ONE. After careful consideration, we feel that it has merit but does not fully meet PLOS ONE’s publication criteria as it currently stands. Therefore, we invite you to submit a revised version of the manuscript that addresses the points raised during the review process.

We look forward to receiving your revised manuscript.

Kind regards,

Anat Gesser-Edelsburg, Ph.D.

Academic Editor

PLOS ONE

Journal Requirements:

“The proposed review will be undertaken as part the main author’s National Institute for Health and Care Research (NIHR) Pre-doctoral Clinical and Practitioner Academic Fellowship (PCAF), reference NIHR304873. “The views expressed are those of the author(s) and not necessarily those of the NIHR or the Department of Health and Social Care” [17]. The main author is sponsored by Homerton Healthcare NHS Foundation Trust.”

“NIHR ICA Pre-doctoral Clinical and Practitioner Academic Fellowship 2024

Reference: NIHR304873”

Reviewers' comments:

Reviewer's Responses to Questions

**Comments to the Author**

1. Does the manuscript provide a valid rationale for the proposed study, with clearly identified and justified research questions?

Reviewer #1: Yes

Reviewer #2: Yes

2. Is the protocol technically sound and planned in a manner that will lead to a meaningful outcome and allow testing the stated hypotheses?

Reviewer #1: Yes

Reviewer #2: Yes

3. Is the methodology feasible and described in sufficient detail to allow the work to be replicable?

Reviewer #1: Yes

Reviewer #2: Yes

4. Have the authors described where all data underlying the findings will be made available when the study is complete?

Reviewer #1: Yes

Reviewer #2: Yes

5. Is the manuscript presented in an intelligible fashion and written in standard English?

Reviewer #1: Yes

Reviewer #2: Yes

You may also provide optional suggestions and comments to authors that they might find helpful in planning their study.

Reviewer #1: Reviewer comments:

• This protocol is on an interesting topic that will no doubt be important to understand and improve the care of the Orthodox Jewish community in the UK. However, upon reviewing the types of articles that PLOS One publishes, it doesn’t seem that a protocol for a scoping review fits what PLOS One is looking for. I am still including my feedback for this paper if the editor decides that it will be accepted.

Introduction:

• Page 3, line 77: you introduce the word “shabbat.” This is the Hebrew term that readers may not be familiar with. You may want to consider introducing it as “the Sabbath” and then in parenthesis, saying (Shabbat/Shabbos).

• You established a good background and gap in the literature and really give the reader a good look into why this work is important. I am wondering if there is any literature you could cite to bolster this regarding whether Orthodox women experience more or high rates of maternity complications compared to the general population in the UK?

Methods:

• The acronym PPI work has been used several times, but it is unclear what it stands for. Please introduce the acronym upon first use, and then use the acronym.

• Grey literature also includes PhD dissertations and thesis, are you including those? If so, be mindful of how long they are for the full-text review and data extraction process.

• Are you planning on using any software, such as Covidence, to house and track your articles and data extraction? If so, please state that in the methods.

• I am worried that given your timeframe of 6 months, you may be biting off more than you can chew by expanding your search to outside of the UK. As you describe in the methods, your research questions are specific to the UK, as your paper title also states. The purpose of a scoping review is to map the data to then guide future work. I understand that your initial search did not turn up many results, but THAT is your gap! Mapping the few studies that have been done will then allow you to discuss the fact that there are many gaps in the literature, and they are x, y, and z based on the findings of your scoping review.

• In scoping reviews, data extraction should first be done separately and independently by 2 people and then compared for consensus. With the shared Excel file, how can you guarantee independent extraction is being done?

Discussion:

• Again, you discuss this review is to identify needs of Orthodox Jewish women in the UK, but if you include literature from Israel, you may get a very different view of needs. Israel has a large Jewish and Orthodox Jewish population, and therefore, many healthcare professionals are likely sensitive to the needs of these patients. In the UK, as you stated, the Orthodox community doesn’t interact much with the external world, so their experience in healthcare situations will likely be different.

In general, I am excited to read the results of your review and learn about your findings.

Reviewer #2: This protocol is timely, well-structured, and clearly justified. The topic, “Exploring the needs and experiences of Orthodox Jewish women and their partners in relation to UK maternity services”, is under-researched and highly relevant for equitable and culturally competent care delivery. The use of Arksey and O’Malley’s framework, PRISMA-ScR guidelines, and the incorporation of PPI (Patient and Public Involvement) enhances the methodological credibility.

Suggestions for improvement:

1. Minor grammar corrections: change “to what extend” to “to what extent”; change “courtiers” to “countries”

2. Consider elaborating on how PPI will directly inform search terms and interpretation.

3. When describing data availability, consider naming a preferred open repository (e.g., OSF, Zenodo) to strengthen transparency.

Overall, this is a robust and valuable protocol that will likely make a meaningful contribution to improving culturally sensitive maternity care. I look forward to the results of the scoping review.

**Do you want your identity to be public for this peer review?** For information about this choice, including consent withdrawal, please see our Privacy Policy

Reviewer #1: No

Reviewer #2: **Yes: ** Ella C. Nelson

---

## [Author Response · Author response to Decision Letter 1]

8 Jul 2025

Reviewer comments:

•

Introduction:

• Page 3, line 77: you introduce the word “shabbat.” This is the Hebrew term that readers may not be familiar with. You may want to consider introducing it as “the Sabbath” and then in parenthesis, saying (Shabbat/Shabbos).

Addressed, please see line 76 on the revised version.

• You established a good background and gap in the literature and really give the reader a good look into why this work is important. I am wondering if there is any literature you could cite to bolster this regarding whether Orthodox women experience more or high rates of maternity complications compared to the general population in the UK?

Comment added, please see lines 91-94

Methods:

• The acronym PPI work has been used several times, but it is unclear what it stands for. Please introduce the acronym upon first use, and then use the acronym. Addressed, please see line 44 on the revised version.

• Grey literature also includes PhD dissertations and thesis, are you including those? If so, be mindful of how long they are for the full-text review and data extraction process.

We planned to search Ethos, however, it has been offline since the cyber attack in October 2023.

• Are you planning on using any software, such as Covidence, to house and track your articles and data extraction? If so, please state that in the methods.

No

• I am worried that given your timeframe of 6 months, you may be biting off more than you can chew by expanding your search to outside of the UK. As you describe in the methods, your research questions are specific to the UK, as your paper title also states. The purpose of a scoping review is to map the data to then guide future work. I understand that your initial search did not turn up many results, but THAT is your gap! Mapping the few studies that have been done will then allow you to discuss the fact that there are many gaps in the literature, and they are x, y, and z based on the findings of your scoping review.

We found no UK study and have decided to amend our review title to reflect that, please see line 3.

• In scoping reviews, data extraction should first be done separately and independently by 2 people and then compared for consensus. With the shared Excel file, how can you guarantee independent extraction is being done?

Addressed, please see lines 214-223

Discussion:

• Again, you discuss this review is to identify needs of Orthodox Jewish women in the UK, but if you include literature from Israel, you may get a very different view of needs. Israel has a large Jewish and Orthodox Jewish population, and therefore, many healthcare professionals are likely sensitive to the needs of these patients. In the UK, as you stated, the Orthodox community doesn’t interact much with the external world, so their experience in healthcare situations will likely be different.

Due to the lack of UK studies, literature from Israel and America will be included, and this will be discussed in the limitations section. We do not expect religious practices to vary between countries.

---

## [Editor Report · Decision Letter 1]

10 Jul 2025

Protocol for a scoping review of the evidence concerning the unique needs and experiences of Orthodox Jewish women and their partners using maternity services in the UK

PONE-D-25-16660R1

Dear Dr. Rivas,

We’re pleased to inform you that your manuscript has been judged scientifically suitable for publication and will be formally accepted for publication once it meets all outstanding technical requirements.

Kind regards,

Anat Gesser-Edelsburg, Ph.D.

Academic Editor

PLOS ONE

Additional Editor Comments (optional):

Kindly ensure that Reviewer 2's minor comments are fully addressed prior to submitting the final version.
---

## [Editor Report · Acceptance letter]

PONE-D-25-16660R1

PLOS ONE

Dear Dr. Rivas,

I'm pleased to inform you that your manuscript has been deemed suitable for publication in PLOS ONE. Congratulations! Your manuscript is now being handed over to our production team.

Kind regards,

on behalf of

Prof. Anat Gesser-Edelsburg

Academic Editor

PLOS ONE